# Taxonomic Identification of Different Species of the Genus Aeromonas by Whole-Genome Sequencing and Use of Their Species-Specific β-Lactamases as Phylogenetic Markers

**DOI:** 10.3390/antibiotics10040354

**Published:** 2021-03-28

**Authors:** Xavier Bertran, Marc Rubio, Laura Gómez, Teresa Llovet, Carme Muñoz, Ferran Navarro, Elisenda Miro

**Affiliations:** 1Institut d’Investigació Biomèdica Sant Pau, 08026 Barcelona, Spain; xavibertran95@gmail.com (X.B.); mrubiobu@santpau.cat (M.R.); 2Department of Microbiology, Hospital de la Santa Creu i Sant Pau, 08041 Barcelona, Spain; LGomezMa@santpau.cat (L.G.); Tllovet@santpau.cat (T.L.); cmunoz@santpau.cat (C.M.)

**Keywords:** MOX, FOX, cephamicinases, beta-lactamases, *Aeromonas dhakensis*, *Aeromonas rivipollensis*, core-genome

## Abstract

Some *Aeromonas* species, potentially pathogenic for humans, are known to express up to three different classes of chromosomal β-lactamases, which may become hyperproduced and cause treatment failure. The aim of this study was to assess the utility of these species-specific β-lactamase genes as phylogenetic markers using whole-genome sequencing data. Core-genome alignments were generated for 36 *Aeromonas* genomes from seven different species and scanned for antimicrobial resistance genes. Core-genome alignment confirmed the MALDI-TOF identification of most of the isolates and re-identified an *A. hydrophila* isolate as *A. dhakensis*. Three (B, C and D) of the four Ambler classes of β-lactamase genes were found in *A. sobria, A. allosacharophila, A. hydrophila* and *A. dhakensis* (*bla*_CphA_*, bla*_AmpC_ and *bla*_OXA_). *A. veronii* only showed class-B- and class-D-like matches (*bla*_CphA_ and *bla*_OXA_), whereas those for *A. media, A. rivipollensis* and *A. caviae* were class C and D (*bla_CMY_*, *bla*_MOX_ and *bla*_OXA427_). The phylogenetic tree derived from concatenated sequences of β-lactamase genes successfully clustered each species. Some isolates also had resistance to sulfonamides, quinolones and aminoglycosides. Whole-genome sequencing proved to be a useful method to identify *Aeromonas* at the species level, which led to the unexpected identification of *A. dhakensis* and *A*.*rivipollensis* and revealed the resistome of each isolate.

## 1. Introduction

The taxonomy of the genus *Aeromonas* is quite complex. The history of this genus of Gammaproteobacteria has been in constant change, as described by Fernandez-Bravo and Figueras [1]. The members of the *Aeromonas* genus are Gram-negative rods, oxidase- and catalase-positive, and glucose fermenters. They are capable of degrading nitrates to nitrites and are resistant to the vibriostatic factor O/129 (2,4-diamino-6,7-di-iso-propylpteridine phosphate) [1]. Despite these established biochemical characteristics, identifying species within the *Aeromonas* genus has always been a convoluted task. Included in the *Vibrionaceae* in 1965, the *Aeromonas* were subsequently classified in their own family, the *Aeromonadaceae*, in 1976 [2]. Since then, 36 different species have been included in the genus, up to 19 of which are considered to be potential human pathogens related to gastrointestinal, as well as blood and wound, infections in both immunocompromised and immunocompetent patients [2]. The most pervasive species are *A. caviae, A. hydrophila, A. veronii* and *A. dhakensis* [2,3,4,5].

Regarding their phenotypical identification, Abbot et al. [6] tested 63 biochemical traits in 193 strains representing 14 different *Aeromonas* species and found that only 15% (9/63) returned consistent results for each species. More elaborate identification methods rely on amplifying housekeeping genes, with *rpo*B*, rpo*D and *gyr*B being the most used, as their interspecies mean sequence similarity in aeromonads is 89% to 92% [2,4,7,8,9]. The analyses can be complemented by other conserved genes, such as *recA, dnaJ, gyrA, dnaX* and *atpD* [7,10], with their intraspecific and interspecific nucleotide differences of <2% and >3%, respectively, serving as baseline values for identifying *Aeromonas* isolates. However, not all the genotypic methods are infallible. De Melo et al. [11] found that an *Aeromonas* isolate was misidentified by multi-locus phylogenetic analysis (MLPA) (*gyrB, rpoD, recA, dnaJ, gyrA, and dnaX*), whereas clustering with reference genomes allowed species identification.

Faster methods such as MALDI-TOF can identify *Aeromonas* to the species level with a relatively low error rate (<10%) [2,12,13], although most databases need curating, as errors are still being reported in clinical practice [14] and many new taxa have been discovered in recent years [15,16]. Accordingly, identification becomes more accurate as new spectra are added to the working databases [15,17]. Nevertheless, identification by MALDI-TOF is not always conclusive: for example, *A*. *dhakensis* has historically been confused with other clinically relevant aeromonads (*A. veronii, A. hydrophila* and *A. caviae*) [2,8,9,10,11,12,13,14,15,16,17,18,19,20]. The phylogenetic study that led to the creation of this taxon was based on the genes *gyr*B and *rpo*D, which were later complemented by *rec*A*, dna*J and *gyr*A in a MLPA [18]. The latest method used to identify different *Aeromonas* strains in clinical settings is whole-genome sequencing (WGS), which has emerged as a promising and highly discriminative tool, not only for epidemiology, but also in taxonomy [21].

*Aeromonas* spp. are traditionally associated with resistance to β-lactams such as ampicillin (with the exception of *A. trota*), as well as other penicillins, and first- and second-generation cephalosporins, whereas they are generally susceptible to third- and fourth-generation cephalosporins, monobactams and carbapenems, as well as aminoglycosides and fluoroquinolones [2,10]. β-lactam resistance in the genus is attributed to the production of two to three chromosomal β-lactamases: an Ambler class B metallo-β-lactamase (MBL), a class C cephalosporinase and a class D penicillinase. In general, *A. allosaccharophila, A. dhakensis, A. hydrophila* and *A. sobria* strains produce all three types of β-lactamase, *A. caviae*, class C and D and *A. veronii*, class B and D [18,20,22,23,24].

Class B MBLs produced by *Aeromonas* mostly belong to the *bla*_CphA_ group and specifically hydrolyse carbapenems. The detection of CphA-like carbapenemases is challenging, requiring a double-disk method combining imipenem or ceftazidime with 500 mM EDTA [4]. Interestingly, the presence of an MBL in aeromonads often does not translate into sufficient carbapenemase activity, even in the presence of inducers [25]. Class C cephalosporinases found in *Aeromonas* belong to the AmpC family and produce a resistance phenotype against cephamycins (cefoxitin and cefotetan), third-generation cephalosporins (cefotaxime, ceftazidime) and β-lactam inhibitors such as clavulanate, tazobactam and sulbactam, but they cannot hydrolyse penicillins or carbapenems. Lastly, Class D penicillinases related to the oxacillinase (OXA) family show high hydrolysis rates for penicillins, low rates for first- and second-generation cephalosporins and do not hydrolyse carbapenems [4,23].

In this study, we used WGS to look for the chromosomal β-lactamase genes carried by different *Aeromonas* spp. and assessed their potential role as species-specific identification genes with core-genome alignment. Additionally, we aimed to elucidate the origin of some CMY, FOX and MOX cephalosporinases described in *Enterobacteriaceae* as an acquired β-lactam resistance mechanism.

## 2. Results

### 2.1. Identification by MALDI-TOF

The MALDI-TOF analysis confirmed the previous routine laboratory identification of 13 *Aeromonas* spp. isolates (Table 1). Both methods correctly identified all the isolates, except for two: the *A*. *allosaccharophila* reference strain (ATCC 51208), which was identified as *A*. *veronii* by MALDI-TOF, and the *A*. *caviae* D547 strain, which was identified as *A. hydrophila* by MALDI-TOF.

### 2.2. WGS Analysis: Core-Genome Alignment

The Roary pangenome analysis of the 22 isolates, 9 type strain genomes and 13 de novo sequenced isolates produced a core-genome alignment of 760,946 bp. The Roary analysis showed 745 core genes (found in > 99% genomes), 44 soft core genes (found in 95–99% genomes), 8011 shell genes (found in 15–95% genomes), and 14,312 cloud genes (found in < 15% genomes) from a total of 23,112 genes.

The core-genome alignment produced eight different clusters representing each of the following species: *A. rivipollensis* (the D180 isolate clustered with the *A. rivipollensis* type strain genome, so we renamed it as a *A*. *rivipollensis* strain), *A. media* (D175), *A. caviae* (D174, D549, D550 and D552), *A. hydrophila* (D173), *A*. *dhakensis* (D547, first identified as *A*. *caviae* by phenotypic methods, and re-identified as *A. hydrophila* by MALDI-TOF), *A. sobria* (D176)*, A. veronii* (D178, D551 and D553) and *A. allosaccharophila* (D179) (Figure 1).

The core-genome sequence analysis allowed us to correctly identify all clinical isolates to the species level. WGS analysis also revealed which *Aeromonas* species share the highest core-genome similarities, thus producing three main phyllogenetic groups: one group with *A. media, A. rivipollensis* and *A. caviae,* a second one with *A. dhakensis* and *A. hydrophila* and a third one with *A. sobria, A. allosaccharophila* and *A. veronii*.

The ANI genome comparison supports the core-genome analysis, and is shown in the Appendix A. Additionally, type strain information is provided in Appendix A.

### 2.3. Antimicrobial Susceptibility Testing

The Antimicrobial Susceptibility Testing (AST) was performed in twelve of the thirteen clinical isolates. *A*. *sobria* D176 was removed from the AST because it did not match the growth standards for the test. Results of AST are shown in Table 2.

Practically all *Aeromonas* isolates studied were resistant to ampicillin, amoxicillin-clavulanate and cefazolin, with three exceptions: *A*. *hydrophila* D173 isolate, which was only resistant to first- and second-generation cephalosporins and *A*. *veronii* (D551), susceptible to amoxicillin-clavulanate (Table 1). Only two strains showed resistance to third-generation cephalosporins and all of them were resistant to carbapenems. We also found resistance to nalidixic acid in three *A. caviae* and two *A. veronii* strains (one of them was also resistant to cotrimoxazol). No species-specific resistance patterns were observed.

### 2.4. β-Lactamases

Given that WGS is a technique not yet available for all clinical microbiology laboratories, and because various β-lactamases have been described in the genus *Aeromonas*, we hypothesized that these enzymes could help achieve a correct identification when standard laboratory techniques are inconclusive.

The assembled genomes obtained from the WGS analysis and the genomes obtained from Genbank were analysed to look for molecular mechanisms associated with β-lactam resistance using the public ResFinder and CARD databases (Table 3). As expected, different classes of β-lactamase genes in the *Aeromonas* genus were found.

The Ambler class B β-lactamase genes detected were mainly of the *bla*_CphA_ family, including those of the *bla*_Imi_-type, and showed very little genetic diversity (Figure 2): *bla*_CphA2,_
*bla*_CphA3,_
*bla*_CphA4,_
*bla*_CphA6_*, bla*_CphA8,_
*bla*_ImiH_ and *bla*_ImiS_. Ambler class C β-lactamase genes were the most diverse, with up to five different families being detected: *bla*_MOX,_ (*bla*_MOX-5,-6,-7,-9_)*, bla*_FOX,_ (*bla*_FOX-2,-7_)*, bla*_AQU-2_ and *bla*_CMY_ (*bla*_CMY-1,_
*bla*_CMY-8b_). Finally, Ambler class D β-lactamase genes belonged to the *bla_OXA_* family, including the *bla*_Amp_ hits (*bla*_OXA-12,-427,-724_ and *bla*_AmpH,_
*bla*_AmpS_).

Thus, the phylogenetic analysis of the β-lactamase genes was carried out by UPGMA and revealed the greatest diversity in the AmpC group (Figure 2), separating the FOX and MOX clusters. This group also possessed the highest species specificity. Following ResFinder nomenclature, all *A. caviae* strains were clustered in the MOX group along with the *A. hydrophila* strains. Nevertheless, *A*. *dhakensis*, which is phylogenetically related to *A*. *hydrophila*, showed a *bla*_CMY-8b_ gene that has 84% identity with *bla*_MOX_ genes. The *A. dhakensis* D547 and the *A. hydrophila* D173 strains expressed the carbapenemases CphA or Imi, which share a similarity of 94.49% and are classified in the same β-lactamase group (Figure 2).

According to the clusters defined by WGS, the first one was composed of *A. sobria, A. allosaccharophila, and A. veronii*, which, despite their proximity did not share the same types of β-lactamase genes. While the *bla_CphA_* and *bla_AMPS_* genes were present in all three species, *bla_FOX_* was not detected among the *A. veronii* strains. 

Finally, the *A*. *media* and *A*. *caviae* strains expressed the cephalosporinases *bla_CMY-8b_* or *bla_MOX-9_* (both with a 96% identity) and the *bla*_OXA-427_ gene, which have between 96.48% and 98.36% similarity with AmpS (Figure 2).

Therefore, although each cluster was characterized by particular β-lactamase genes, the results did not reveal a sufficiently clear pattern to establish an algorithm for the identification of *Aeromonas* spp. It would also be important to unify a single nomenclature between the different databases consulted.

### 2.5. Characterisation of Other Mechanisms of Resistance to Antimicrobials 

Aside from the expected β-lactam resistances, five of the 13 isolates (two *A. veronii*: D551 and D553, and three *A. caviae* D549, D550 and D553) also showed visible resistance to nalidixic acid. The nalidixic acid resistance was associated with substitutions identified in the *gyr*A gene in strains D550 and D551 (Ser83Ile) and strain D553 (Ser83Arg). In strains D549 and D552, a substitution was detected in the *gyrA* gene (Ser83Ile) and in the *parC* gene (Ser80Ile). No mutations producing resistance were found in *gyr**B* or *parE*, and the mutations detected were not associated with an increased quinolone resistance.

Regarding resistances to aminoglycosides, although this drug family was not included in the AST, two of the isolates expressed aminoglycoside-modifying enzymes (AME): strain D549 carried the genes *aph*(3′′)*-*1b and *aph(*3′′*)-*1d, which have the highest affinity for streptomycin, whereas strain D552 possessed *aadA2*, which hydrolyses streptomycin and spectinomycin.

Only one *A*. *veronii* strain (D551) resistant to trimethoprim-sulfamethoxazole carried the *sul*1 and *dfrA*12 genes. Query covers and identities for both genes were 100%.

## 3. Discussion

At present, identification of *Aeromonas* isolates to the species level remains a complex task [6,11,14]. Nevertheless, we obtained results of high accuracy using a WGS method and core-genome alignment, as have other studies [21,26,27]. This promising technique, however, is out of reach for routine use in the laboratory, and the most commonly used method, MALDI-TOF, may misidentify samples without a sufficiently comprehensive database [13,16], as occurred in our *A*. *allosaccharophila*, *A*. *dhakensis* and *A. rivipollensis* strains. *A*. *rivipollensis* was described for the first time in 2015, also in Catalonia, from the Ter river and, in agreement with our results, the authors describe it as a new species related with *A*. *media* [28]. To improve identification in *Aeromonas*, we investigated the potential of AmpC β-lactamase genes to serve as species-specific markers and, in our experience, the PCR amplification and Sanger sequencing analysis of the AmpC β-lactamase class can improve the correct identification to the species level in the genus *Aeromonas.*

*Aeromonas* spp. are known to carry several chromosomal β-lactam resistance genes belonging to Ambler classes B, C and D [18,20,22,23,24]. By obtaining the genetic sequences of these β-lactamase genes through WGS, we hoped to shed light on whether the class C cephalosporinases MOX and FOX, now pervasive among the *Enterobacteriaceae*, originate from the *Aeromonas* genus. Ebmeyer et al. [29] detected different MOX-type enzymes in the genome of various species of *Aeromonas*, including *A*. *sanaralli* (MOX-1), *A*. *caviae* (MOX-2) and *A*. *media* (MOX-9). The same authors have also reported that the origin of FOX enzymes could be in the chromosome of *A*. *allosaccharophila* [30].

The high-resolution method of WGS allowed almost every *Aeromonas* species to be matched with a sequence type genome. The phylogenetic tree confirmed that *A*. *sobria* and *A*. *allosaccharophila* differ considerably from the genetically closest species *A. veronii* in the core-genome aligment, as previously described by MLPA or by *gyr*B and *rpo*B phylogenetic analysis [7,8]. The WGS approach also led to the unexpected identification of an *A. dhakensis* isolate, originally identified as *A. hydrophila* by MALDI-TOF, a species not previously reported as a clinical strain in Spain. WGS also identified an *A. rivipollensis* isolate previously identified as *A. media*.

Using WGS analysis, we were able to determine the resistance genes carried by the *Aeromonas* isolates. Three types of β-lactamase genes (*bla*_CphA_, *bla*_AmpC_ and *bla*_OXA_) were found in the isolates, in accordance with the species-specific patterns described in the literature. *A. sobria, A. allosacharophila, A hydrophila* and *A. dhakensis* carried all three types, *A. veronii* was the only clinical strain to possess only *bla*_CphA_ and *bla*_OXA_, and both *A. media*, *A. caviae* and *A*. *rivipollensis* carried a *bla*_AmpC_ and a *bla*_OXA_ gene [18,20,22,23,24].

It has already been reported that there is no agreement between the presence of so many genes encoding different β-lactamases (which should confer resistance to most β-lactam antibiotics) with the actual resistance profile. A two-component regulator (TCR), closely related to the CreBC of *Escherichia coli*, has been described in *Aeromonas*. This TCR includes a putative mutant form of a transcription factor, the BlrA protein (related to the extended family of phosphorylation-dependent response regulators), whose gene was found immediately upstream from the *blr*B gene, encoding a predicted sensor kinase [25,31]. The presence of this operon prevents the expression of β-lactamase genes, and mutations in this system confer a high level of resistance to β-lactams. In some *Aeromonas* species (*A*. *veronii*, *A*. *hydrophila* and *A*. *caviae*), a frequency range of *blr*AB de-repression between 10^7^ and 10^9^ has been described, which increases the MICs of the β-lactams tested by 16 for ampicillin, 4 for imipenem, up to 16 for cephaloridine and up to 129 for cefotaxime [32].

Notably, the Ambler class C β-lactamases showed a pattern of high species-specificity, indicating that each *Aeromonas* species has a characteristic family of cephalosporinase genes (if any): *bla*_FOX,_ in *A. sobria, A. allosacharophila and A. hydrophila, bla*_AQU_ in *A. dhakensis, bla*_CMY/MOX-9_ in *A. media* and *A. rivipollensis,* and *bla*_MOX_ in *A. caviae.* Therefore, plasmid-mediated *AmpC* genes are derived from the chromosomal *AmpC* genes of several members of the family *Enterobacteriaceae*, including *Enterobacter cloacae* (MIR o ACT group), *Morganella morganii* (DHA group), and *Hafnia alvei* (ACC group) [33]. As reported by Jacoby et al. [34], CMY is represented twice, as it has two quite different origins. Six current varieties (CMY-1,-8,-9,-10,-11, and -19) are related to chromosomally determined AmpC genes in *Aeromonas* spp., while the remainder are related to AmpC β-lactamases of *Citrobacter freundii*.

The chromosomal β-lactamase genes of *A*. *caviae* thus seem to be the progenitors of plasmid-mediated MOX-2 and MOX-4 and of some CMY-1-related enzymes. The plasmid-mediated FOX enzymes seem to have their origin in the AmpC CAV-1 of *A*. *media* (first identified as *A*. *punctata*) [35]. Finally, *A*. *jandaei* and *A*. *enteropelogenes* also have their own chromosomal AmpC β-lactamase genes (AsbA1 and TRU-1, respectively) [4,36]. These data support that each *Aeromonas* species has its own chromosomal β-lactamase genes and is a potential progenitor of new emerging *AmpC* enzymes.

As it is mandatory to use more than one public resistance database, we found that, using the ResFinder and CARD, some entries were exclusive to either one or the other: *bla*_AmpS_ and *bla*_AmpH_ only appeared in ResFinder and *bla*_AQU-1_, *bla*_AQU-2_, *bla*_MOX-9_ and *bla*_CepS_ were exclusive to CARD. These specific entries yielded matches with higher identities for our query sequences, indicating that completeness is essential for proper diagnosis and that more than one curated database for antimicrobial resistance genes should be used when screening isolates [37]. Nevertheless, regarding *Aeromonas* resistance genes, both ResFinder and CARD are good tools for screening, but we found that the percentage of identity for many species-specific resistance genes, such as *bla*_AmpC_ of *A. hydrophila* and *bla*_OXA_-like of *A. caviae*, was too low.

Cases of fluoroquinolone resistance mediated by genetic mobile elements have been previously described in *Aeromonas* [4,10,27,38,39]. However, we did not find any *qnr*-like genes among the resistant isolates D549-D553, correlating with the findings of Ghatak et al. [27]. Additionally, we found D549-D552 to have point mutations in *gyr*A resulting in amino acid substitution (Ser83Ile) and *par*C (Ser80Ile), which are typically associated with quinolone-resistant aeromonads [40,41], as well as a mutation in *gyrA*: Ser83Arg in strain D553 not previously reported in *Aeromonas* spp.

The results for trimetoprim-sulfamethoxazole AST correlate with findings by Kadlec et al. [42], who reported that 100% (33/33) of *Aeromonas* spp. carrying *sul1* exhibited increased MICs for sulfamethoxazole treatments. Aminoglycoside resistance genes found in strain D549 [*aph(3*′′*)1b* and *aph(3*′′*)1d*)] match previous reports on *Aeromonas* [24,40], whereas we were unable to find publications mentioning *aadA2,* as we found in strain D552.

Overall, WGS allowed the successful identification of all the *Aeromonas* isolates to the species level. β-lactamase genes, naturally associated with each species of the genus (a class B carbapenemase, a class C cephalosporinase and a class D penicillinase), were present in our isolates in different combinations: *A. sobria, A. allosacharophila, A hydrophila* and *A. dhakensis* carried all three types, *A. veronii* was the sole clinical strain to possess only a *bla*_CphA_-type and a *bla*_OXA_-type gene, whereas *A. media, A. rivipollensis* and *A. caviae* carried a *bla*_AmpC_ and a *bla*_OXA_ gene. To our knowledge, this is the first report in Spain of an *A. dhakensis* isolate, previously identified as *A. hydrophila* by MALDI-TOF. 

Based on the results obtained, we can conclude that WGS is the most appropriate technique both to identify *Aeromonas* at the species level and to describe the different resistance mechanisms present. In addition, this method produces objective results that are comparable with those from any other laboratory anywhere. The only drawback for its laboratory application, at least at present, is the high cost.

## 4. Materials and Methods 

### 4.1. Identification and Antimicrobial Susceptibility Testing

Thirteen strains were included in this study. Seven (D173–D176, D178–D180) were from the Colección Española de Cultivos Tipo (CECT) and six (547, 549–553) were isolated in our laboratory from faeces of patients with gastroenteritis. Strains were stored at −20 °C until use, and one copy at −80 °C.

Strains were grown in blood agar medium and underwent MALDI-TOF MS Autoflex II (Bruker Daltonics, Barcelona, Spain) identification from single colonies. From the same cultures, 2–3 colonies were used to prepare a 0.5 McFarland suspension [43,44], which was used in disk-diffusion susceptibility tests for Gram-negative *Enterobacteriaceae* in Mueller-Hinton agar. Cultures were incubated at 37 °C for 18–24 h. The antimicrobial drugs used were: ampicillin (10 μg), amoxicillin-clavulanic acid (30 μg), piperacillin (30 μg), cefazolin (30 μg), piperacillin-tazobactam (36 μg), cefoxitin (30 μg), cefotaxime (5 μg), ceftazidime (10 μg), aztreonam (30 μg), cefepime (30 μg), ertapenem (10 μg), imipenem (10 μg), cefuroxime (30 μg), ciprofloxacin (5 μg), nalidixic acid (30 μg) and trimethoprim/sulfamethoxazole (25 μg). Cut-off thresholds were established according to the EUCAST criteria [43], as EUCAST defines clinically relevant *Aeromonas* spp. (*A. hydrophila, A. veronii, A. dhakensis and A. caviae*) as intrinsically resistant to ampicillin, amoxicillin, amoxicillin-clavulanic acid, ampicillin-sulbactam and cefoxitin [45].

### 4.2. DNA Extraction and Whole-Genome Sequencing

From single colonies grown in blood agar plates, the thirteen samples were transferred to an enriched growth broth (5 mL Luria Bertani) and cultured overnight at 37 °C. DNA extraction and purification was done according to the DNeasy^®^ UltraClean^®^ Microbial Kit protocol (Qiagen, Frederick, MD, USA).

The quality screening for DNA extraction was done by Invitrogen QUBIT^®^ 3.0 fluorometer, which determines the amount of extracted DNA (ng/μL) per sample. Concentrations that we deemed too low were concentrated using the SpeedVac at 45 °C for 10 min, thus lowering the final volume from 50 μL to 30 μL, and analyzed again. Spectrophotometry at 260 and 280 nm was also calculated to perform a quality control for DNA concentrations and purity using Nanodrop 2000/2000c(Thermofisher Scientific, Waltham, MA, USA).

Sequencing was performed by Sequentia Biotech S.L. (Barcelona, Spain) using Illumina technologies. The sequencing libraries were prepared using a Nextera XT kit (Illumina, Madrid, Spain) and sequenced on an Illumina MiSeq sequencer which generated 2 × 250 bp paired-end reads.

### 4.3. Computational Analysis

Trim_galore v0.6.5 (available at https://github.com/FelixKrueger/TrimGalore/tree/ accessed on 1 January 2021, Babraham Bioinformatics, Cambridge, UK, 2019) was used to trim Illumina adaptors, and filter low-quality reads and reads under 50 bp. Then Unicycler v0.4.9b (available at github.com/rrwick/Unicycler, Ryan Wick, Victoria, Australia, 2019) was used as a SPAdes v3.14.0 (available at https://cab.spbu.ru/software/spades/ accessed on 1 January 2021, Center for Algorithmic Biotechnology, St Petersburg, Russia, 2019) optimizer in order to generate the best possible assemblies with the Illumina reads. Afterwards, we annotated the genome assemblies using Prokka v1.14.6 (available at https://github.com/tseemann/prokka accessed on 1 January 2021, Torsten Seemann, Victoria, Australia, 2020) [46]. Prokka gff files output was used as input in Roary v1.7.7 (available at https://github.com/sanger-pathogens/Roary accessed on 1 January 2021, Sanger-Pathogens, Cambridge, UK, 2019) [47]. Gubbins v2.4.1 (available at https://github.com/sanger-pathogens/gubbins accessed on 1 January 2021, Sanger-Pathogens, Cambridge, UK, 2020) [48] was used to remove possible recombination events in the core genome alignment. RaxML v8.2.12 (available at https://cme.h-its.org/exelixis/web/software/raxml/ accessed on 1 January 2021, The Exelixis Lab, Heidelberg, Germany, 2018) was used with the core-genome alignment file to infer phylogeny, applying a General Time Reversible (GTRACAT) model with 99 bootstraps.

#### 4.3.1. Whole-Genome Comparison

Complete genomes of *Aeromonas* were downloaded from GenBank (Table 3). Type strain genomes for ANI genome comparison and core genome analysis were downloaded from the NCBI Assembly database (available at https://www.ncbi.nlm.nih.gov/assembly/) (accessed on 9 February 2021). We used OrthoANIu standalone tool for ANI genome comparison (available at https://www.ezbiocloud.net/tools/orthoaniu) Not applicable) [49].

#### 4.3.2. Detection of Antimicrobial Resistance Genes 

The antimicrobial resistance genes in the strains were detected using the CARD [50], ResFinder [51] and complete NCBI (Genbank) [52] databases with the software Abricate (available at https://github.com/tseemann/abricate) (accessed on 17 April 2020). Hits below 60% identity were automatically discarded. 

#### 4.3.3. Antimicrobial Resistance Genes Phylogenetic Tree

Nucleotide sequences of the genes listed in Table 2 searched against the CARD and ResFinder databases were downloaded as FASTA files. We used BLAST to extract the β-lactamase genes from the genomes of our strains. MEGA X [53] was used for FASTA alignment and UPGMA tree calculation and visualization. All genes used for this analysis are indicated in Appendix A.

## Figures and Tables

**Figure 1 antibiotics-10-00354-f001:**
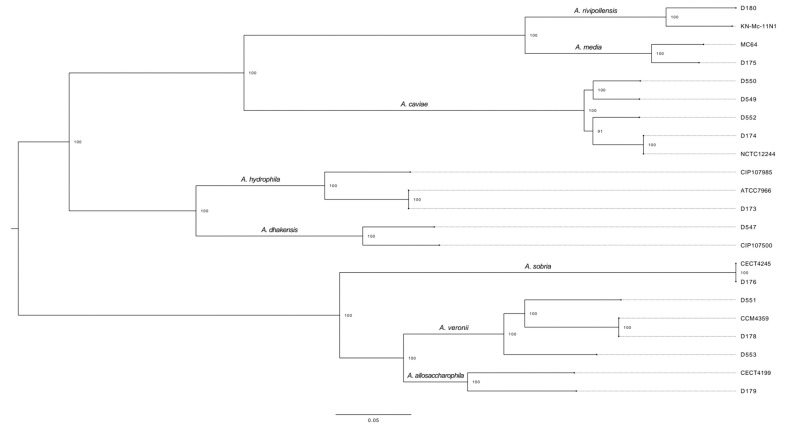
Maximum likelihood phylogenetic tree derived from the core-genome alignment of the *Aeromonas* spp. isolates and type strain genomes. All strains with names beginning with D are isolates from the laboratory. Bootstrap values above 80% are shown in branch nodes (premutation = 100). The different clusters within the phylogenetic tree contain previously identified sequences posted on GenBank. Type strain information and accession numbers are presented in Appendix A.

**Figure 2 antibiotics-10-00354-f002:**
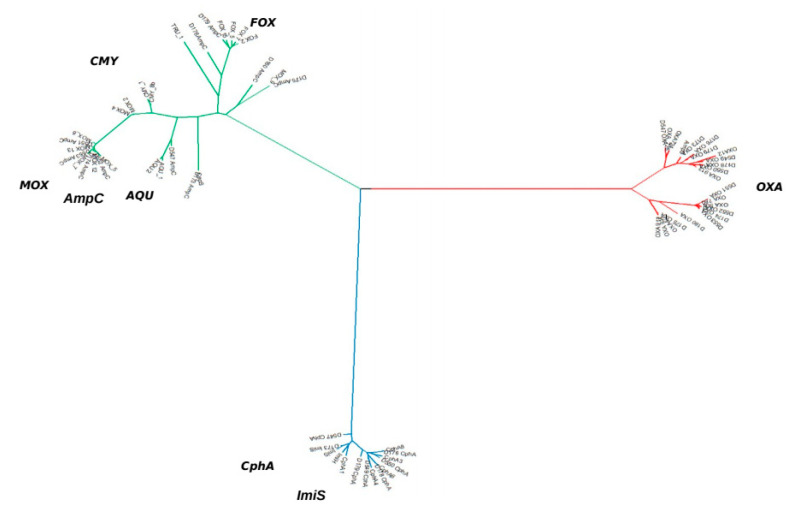
UPGMA tree containing all β-lactamase resistance genes found in *Aeromonas* isolates. Class B β-lactamase branch is coloured in blue, Class C β-lactamase branch is coloured in green, and Class D β-lactamase branch is coloured in red. Analyses of the 65 nucleotide sequences were performed using MEGA X.

**Table 1 antibiotics-10-00354-t001:** MALDI-TOF identification of the 13 *Aeromonas* isolates previously identified in routine laboratory work.

LAB ID	Routine Laboratory Identification	MALDI-TOF Identification	Score
D180 **	*A. media*	*A. media* DSM 4881T HAM	2.22
D175	*A. media* ATCC 33907 (CECT4232)	*A. media* DSM 4881T HAM	2.38
D174	*A*. *caviae* ATCC 15468 (CECT838)	*A. caviae* CECT838T DSM	2.40
D549	*A. caviae*	*A. caviae* CECT838T DSM	2.43
D550	*A. caviae*	*A. caviae* CECT838T DSM	2.41
D552	*A. caviae*	*A. caviae* CECT838T DSM	2.31
D547 *	*A. caviae*	*A. hydrophila* CECT 839T DSM	2.34
D173	*A. hydrophila* ATCC 7966 (CECT839)	*A. hydrophila* CECT 839T DSM	2.47
D176	*A. sobria* ATCC 43979 (CECT4245)	*A. sobria CECT 4245T DSM*	2.45
D178	*A. veronii* ATCC 35624 (CECT4257)	*A. veronii* CECT5761T DSM	2.26
D551	*A. veronii*	*A. veronii* DSM 7386T HAM	2.38
D553	*A. veronii*	*A. veronii* CECT4257T DSM	2.30
D179	*A. allosaccharophila* ATCC 51208 (CECT4911)	*A. veronii* DSM 11576THAM	2.28

Identified as * *A*. *dhakensis* and ** A. rivipollensis by WGS (see below).

**Table 2 antibiotics-10-00354-t002:** Antimicrobial susceptibility test results for the 13 *Aeromonas* strains.

LAB ID	WGS Identification	AMP	AMC	PRL	TPZ	CFZ	FOX	CXM	CTX	CAZ	ATM	FEP	IMP	ETP	NAL	CIP	SXT
D180	*A. rivipollensis*	**6**	**7**	24	24	**14**	33	38	30	37	46	42	32	32	33	36	29
D175	*A. media*	**6**	**12**	22	25	**7**	16	26	30	29	S	36	23	27	31	34	19
D174	*A*. *caviae*	**6**	**12**	22	26	**9**	17	28	35	31	40	36	23	24	31	37	22
D549	*A. caviae*	**6**	**16**	25	31	19	33	38	36	34	45	40	27	30	**6**	34	29
D550	*A. caviae*	**6**	**17**	27	26	28	34	38	39	35	42	40	21	26	**6**	26	S
D552	*A. caviae*	**6**	**12**	28	28	**7**	17	24	32	26	42	36	30	22	**6**	27	22
D173	*A. hydrophila*	27	30	29	32	**6**	**11**	30	29	28	36	34	34	32	32	30	28
D547	*A. dhakensis*	**6**	**11**	23	31	**6**	**6**	25	**16**	31	52	40	25	27	34	39	34
D178	*A. veronii*	**6**	**12**	20	23	15	27	30	30	33	34	32	18	22	31	37	22
D551	*A. veronii*	**6**	26	32	32	**13**	29	36	36	36	44	40	35	S	**6**	29	**6**
D553	*A. veronii*	**6**	**15**	25	28	**8**	22	34	38	32	42	40	28	22	**6**	31	22
D179	*A. allosaccharophila*	**6**	**16**	20	23	**6**	20	32	6	32	44	36	21	23	34	40	21

AMP: Ampicillin; AMC: Amoxicillin-Clavulanate; PRL: Piperacillin; CFZ: Cefazolin; TPZ: Tazobactam-Piperacillin; FOX: Cefoxitin; CXM: Cefuroxime; CTX: Ceftazidime; CAZ: Cefazoline; ATM: Aztreonam; FEP: Cefepime; IMP: Imipinem; ETP: Ertapenem; CIP: Ciprofloxacin; NAL: Nalidixic Acid; SXT: Cotrimoxazol. The numbers represent the diameter of the halos in millimeters. Resistances are in bold. Strain D176 did not grow sufficiently to have readable halos.

**Table 3 antibiotics-10-00354-t003:** Comparison of ResFinder and CARD databases to screen for β-lactamase genes.

Species	ResFinder (% ID)			CARD (% ID)		
	Class B	Class C	Class D	Class B	Class C	Class D
*A. rivipollensis* (*n* = 1)		*bla*_CMY-1_ (95.21) *bla*_CMY-8b_ (94.87)	*bla*_OXA-427_ (100)		*bla*_MOX-9_ (99.74–99.48)	*bla*_OXA-427_ (100)
*A. media* (*n* = 2)		*bla*_CMY-8b_ (96.26)	*bla*_OXA-427_ (100–99.12)		*bla*_MOX-9_ (100)	*bla*_OXA-427_ (100–99.12)
*A. caviae* (*n* = 5)		*bla*_MOX-5_ (99.91) *bla*_MOX-6_ (99.91) *bla*_MOX-7_ (100) (*n* = 3)	*bla*_OXA-427_ (100–98.87)		*bla*_MOX-5_ (99.91) *bla*_MOX-6_ (99.91) *bla*_MOX-7_ (100) (*n* = 3)	*bla*_OXA-427_ (100–98.87)
*A. hydrophila* (*n* = 2)	*bla*_ImiS_ (99.61)	*bla*_MOX-5_ (91.67)	*bla*_AmpS_ (100)	*bla*_ImiS_ (99.61)	*bla*_CepS_ (100)	*bla*_OXA-12_ (98.74)
*A. dhakensis* (*n* = 2)	*bla*_CphA2_ (100) *bla*_ImiH_ (99.74)	*bla*_CMY-8b_ (91.99–98.35)	*bla*_AmpH_ (100)	*bla*_CphA2_ (100) *bla*_ImiH_ (99.74)	*bla*_AQU-2_ (100-93.7)	*bla*_OXA-724_ (100)
*A. sobria* (*n* = 2)	*bla*_CphA8_ (100)	*bla*_FOX-2_ (98.09)	*bla*_AmpS_ (98.87)	*bla*_CphA8_ (100)	*bla*_FOX-2_ (98.09)	*bla*_OXA-12_ (97.48)
*A. veronii* (*n* = 4)	*bla*_CphA3_ (100) *bla*_CphA4_ (99.48) *bla*_CphA6_ (99.87) (*n* = 2)		*bla*_AmpS_ (100)	*bla*_CphA3_ (100) *bla*_CphA4_ (99.48) *bla*_CphA6_ (99.87) (*n* = 2)		*bla*_OXA-12_ (99.12)
*A. allosacharophila* (*n* = 2)	*bla*_CphA4_ (99.48)	*bla*_FOX-7_ (100) *bla*_FOX-2_ (100)	*bla*_AmpS_ (98.87)	*bla*_CphA4_ (99.48)	*bla*_FOX-7_ (100) *bla*_FOX-2_ (100)	*bla*_OXA-12_ (98.99–98.74)

## Data Availability

All WGS data was uploaded to the SRA under the PRJNA685948 Bioproject. Assembled and annotated genome data are available at genbank under accession numbers: JAGDEM000000000, JAGDEN000000000, JAGDEO000000000, JAGDEP000000000, JAGDEQ000000000, JAGDER000000000, JAGDES000000000, JAGDET000000000, JAGDEU000000000, JAGDEV000000000, JAGDEW000000000, JAGDEX000000000, and JAGDEY000000000.

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
