# Peer review of "Taxonomic Identification of Different Species of the Genus Aeromonas by Whole-Genome Sequencing and Use of Their Species-Specific β-Lactamases as Phylogenetic Markers"

_antibiotics, 2021, doi:10.3390/antibiotics10040354_

Round 1

Reviewer 1 Report

The authors of this paper presented results of solid genomic studies on Aeromonas, with special emphasis to analyze genes coding for beta-lactamases. Although general conclusios are valid, presentation must be significantly improved.

  1. The title is misleading. It suggests that mechanisms of resistance of bacteria to antibiotics were studied which is completely not true. Moreover, only genes coding for beta-lactamases were analyzed in details, thus, this fact should be also reflected in the title.
  2. The authors are very imprecise and unstable throughout the text for genetic/biochemical nomenclature. Though only genes coding for beta-lactamases were investigated, sometimes the text indicates actually what was studied but in other places, it appears like enzymes were investigaed which is not true. The authors MUST be consistent throughout the text and indicate clearly that no biochemical characterization of beta-lactamases was performed, and only corresponding genes were analyzed. This can be simply achieved by indicating that the authors are talking about genes, wherever beta-lactamase in mentioned. The best example of confusion can be find just in the first sentence of the abstract. In lines 12-13, it is stated: "... are known to express up to three different classes of chromosomal beta-lactamases". I agree that using such a jargon we can understand what did authors mean, however, precise and formally accurate language must be used in a scientific article. In the current form, this sentence does not make sense. First, genes are expressed, not proteins (proteins are genes' products, and can be synthesized or produced, but not expressed). Second, beta-lactamases are enzymes, thus, there are no "chromosomal beta-bactamases". One can say about corresponding chromosomal genes or chromosomally-encoded beta-lactamases, but not "chromosomal beta-lactamases", as the latter term would indicate that beta-lactamases are proteins which are part of chromosomes or interact with them physically. This is just an example, and the text is full of such mistakes which MUST be corrected.
  3. Line 79: Apart from the same problem as indicated in point 2, I was misled, as after reading this sentence I though that biochemical characterization of beta-lactamases (studied as proteins) was performed. This was not the case, as only genetic analyzes were presented. As stated in point 2, this problem MUST be solved throughout the whole manuscript.
  4. A Table with the list of used Aeromonas strains should be cited in section 2.1. Currently, this is Table 3, but it should appear earlier in the manuscript, perhaps as Table 1 if Results are presented before Materials and Methods.
  5. Line 93: Be sure to use italic font for names of species and genera.
  6. Line 109: "twelven" - did the authors mean "twelve"?
  7. Table 1. What do the presented numbers mean? Neither headings of columns nor footnotes inrom what values are presented in the main body of the table. Are there growth inhibition zones (if so, in what units?) or MIC values (if so, MIC(50) or MIC(90)?) or other values?

Generally, this paper contains solid data, but presentation of the study MUST be intensively corrected.

Author Response

Reviewer 1

The authors of this paper presented results of solid genomic studies on Aeromonas, with special emphasis to analyze genes coding for beta-lactamases. Although general conclusios are valid, presentation must be significantly improved.

  1. The title is misleading. It suggests that mechanisms of resistance of bacteria to antibiotics were studied which is completely not true. Moreover, only genes coding for beta-lactamases were analyzed in details, thus, this fact should be also reflected in the title.

We agree with the reviewer. The title may not express well our two objectives: to correctly identify the different species of Aeromonas using WGS, and to use the β-lactamases present in this genus as possible housekeeping genes. 

Following the reviewer's suggestion, we have changed the title to: “Taxonomic identification of different species of the genus Aeromonas by Whole-Genome Sequencing and use of their species-specific β-lactamases as phylogenetic markers”

  1. The authors are very imprecise and unstable throughout the text for genetic/biochemical nomenclature. Though only genes coding for beta-lactamases were investigated, sometimes the text indicates actually what was studied but in other places, it appears like enzymes were investigaed which is not true. The authors MUST be consistent throughout the text and indicate clearly that no biochemical characterization of beta-lactamases was performed, and only corresponding genes were analyzed. This can be simply achieved by indicating that the authors are talking about genes, wherever beta-lactamase in mentioned. The best example of confusion can be find just in the first sentence of the abstract. In lines 12-13, it is stated: "... are known to express up to three different classes of chromosomal beta-lactamases". I agree that using such a jargon we can understand what did authors mean, however, precise and formally accurate language must be used in a scientific article. In the current form, this sentence does not make sense. First, genes are expressed, not proteins (proteins are genes' products, and can be synthesized or produced, but not expressed). Second, beta-lactamases are enzymes, thus, there are no "chromosomal beta-bactamases". One can say about corresponding chromosomal genes or chromosomally-encoded beta-lactamases, but not "chromosomal beta-lactamases", as the latter term would indicate that beta-lactamases are proteins which are part of chromosomes or interact with them physically. This is just an example, and the text is full of such mistakes which MUST be corrected.

The manuscript has been revised and corrected as suggested by the reviewer.

  1. Line 79: Apart from the same problem as indicated in point 2, I was misled, as after reading this sentence I though that biochemical characterization of beta-lactamases (studied as proteins) was performed. This was not the case, as only genetic analyzes were presented. As stated in point 2, this problem MUST be solved throughout the whole manuscript.

The paragraph has been adapted according to the suggestion, using nomenclature for proteins, because we are writing about the beta-lactamase phenotype.

  1. A Table with the list of used Aeromonas strains should be cited in section 2.1. Currently, this is Table 3, but it should appear earlier in the manuscript, perhaps as Table 1 if Results are presented before Materials and Methods.

Table 1 has been added in section 2.1 and subsequently all the table numbers have been changed.

  1. Line 93: Be sure to use italic font for names of species and genera.

The text has been checked and now italics are used for all the genus and species names.

  1. Line 109: "twelven" - did the authors mean "twelve"?

Yes, the error has been corrected.

  1. Table 1. What do the presented numbers mean? Neither headings of columns nor footnotes inrom what values are presented in the main body of the table. Are there growth inhibition zones (if so, in what units?) or MIC values (if so, MIC(50) or MIC(90)?) or other values?

A text has been added in the foot notes of Table 1 (renumbered as Table 2), explaining that the numbers represent the diameter of the halos in millimetres.

Generally, this paper contains solid data, but presentation of the study MUST be intensively corrected

We hope that the changes done in response to both reviewers have improved the presentation.

Reviewer 2 Report

Bertran X et al. “Taxonomic identification and characterization of antimicrobial resistance mechanisms in the genus Aeromonas by Whole-Genome Sequencing” (Antibiotics-1082061)

In this paper, the authors verified the accuracy of species identification and confirmed the antibiotic resistance gene by analyzing data obtained from the Aeromonas strain and public database that directly performed whole genome sequencing. Through this, a case where species identification through MALDI-Tof analysis was incorrect was identified, and resistance genes were also found. However, what I would like to point out as the biggest weakness of this paper is that there is no discussion about the relationship between the resistance genes identified through this method and actual resistance. The authors discussed about the relationship between the pattern of the presence of a resistance gene and species clustering, but the more important thing in relation to antibiotic resistance is whether the presence of the gene or the mutation are related to the actual antibiotic resistance. In addition, it is needed to discuss the value of identifying the Aeromonas species through WGS.

Line 275. It is not known which strain was directly subjected to whole genome sequencing.

Table 1. I don’t know what units for the numbers in the Table mean.

Author Response

Reviewer 2

In this paper, the authors verified the accuracy of species identification and confirmed the antibiotic resistance gene by analyzing data obtained from the Aeromonas strain and public database that directly performed whole genome sequencing. Through this, a case where species identification through MALDI-Tof analysis was incorrect was identified, and resistance genes were also found. However, what I would like to point out as the biggest weakness of this paper is that there is no discussion about the relationship between the resistance genes identified through this method and actual resistance. The authors discussed about the relationship between the pattern of the presence of a resistance gene and species clustering, but the more important thing in relation to antibiotic resistance is whether the presence of the gene or the mutation are related to the actual antibiotic resistance. In addition, it is needed to discuss the value of identifying the Aeromonas species through WGS.

The reviewer is completely correct in pointing out the lack of discussion about the relationship between the resistance genes identified by this method and the actual resistance. Accordingly, we have added a paragraph in the discussion section: lines 216-226. “It has already been described that there is no agreement between the presence of so many genes encoding different beta-lactamases (which should confer resistance to most beta-lactam antibiotics), with the actual resistance profile. A two-component regulator (TCR), closely related to the CreBC of Escherichia coli, has been described in Aeromonas. This TCR includes a putative mutant form of a transcription factor, the BlrA (related to the extended family of phosphorylation-dependent response regulators), whose gene was found immediately upstream from the blrB gene, encoding a predicted sensor kinase (25, 28). The presence of this operon prevents the expression of beta-lactamases genes, and mutations in this system confer a high level of resistance to beta-lactams. In some Aeromonas species (A. veronii, A. hydrophila and A. caviae), a frequency range of blrAB de-repression between 10 7 and  109 has been described, which increases the MICs of the β-lactams tested by 16 for ampicillin, 4 for imipenem, up to 16 for cephaloridine and up to 129 for cefotaxime (29).”

In addition ti lines 195: “we obtained results of high accuracy using WGS method and core-genome alignment, as have other studies [21, 26-27].”  And line 209: “The WGS approach also led to the unexpected identification of an A. dhakensis isolate,” A final paragraph to discuss the importance of WGS has been added at the end of the discussion section. Lines: 269-272.

Line 275. It is not known which strain was directly subjected to whole genome sequencing.

The number of samples subjected to WGS has been included.

Table 1. I don’t know what units for the numbers in the Table mean.

A text has been added in the foot notes of Table 1 (renumbered as Table 2) explaining that the numbers represent the diameter of the halos in millimetres.

Round 2

Reviewer 1 Report

The authors improved the presentation according to previous comments. I do not have further critical points. 

Author Response

Thank you.

Reviewer 2 Report

.

Author Response

Thank you.